# Health system use among patients with mental health conditions in a community based sample in Toronto, Canada: A retrospective cohort study

Kimberly Lazare[1,2]*, Sumeet Kalia[3,4], Babak Aliarzadeh[3,4], Steven Bernard[2],
Rahim Moineddin[1], David Eisen[1,2], Michelle Greiver[1,2,3], David Kaplan[1,2],
David Koczerginski[2,5], Maria Muraca[1,2], Wai Lun Alan Fung[2,5], Braden O'Neill[2,6]

1 Department of Family & Community Medicine, University of Toronto, Toronto, Canada, 2 North York
General Hospital, Toronto, Canada, 3 University of Toronto Practice-Based Research Network, University of
Toronto, Toronto, Canada, 4 Research and Innovation, North York General Hospital, Toronto, Canada,
5 Department of Psychiatry, University of Toronto, Toronto, Canada, 6 Unity Health, Toronto, Canada

* Kimberly.Lazare@nygh.on.ca

pone.0266377

ALBANIA

**Data Availability Statement:** The data that was
used in this study is available from the Office of
Research and Innovation at North York General

## Abstract

### Objective

To identify hospital and primary care health service use among people with mental health
conditions or addictions in an integrated primary-secondary care database in Toronto,
Ontario.

### Method

This was a retrospective cohort study of adults with mental health diagnoses using data
from the Health Databank Collaborative (HDC), a primary care-hospital linked database in
Toronto. Data were included up to March 31st 2019. Negative binomial and logistic regression were used to evaluate associations between health care utilization and various patient
characteristics and mental health diagnoses.

### Results

28,482 patients age 18 or older were included. The adjusted odds of at least one mental
health diagnosis were higher among younger patients (18–30 years vs. 81+years aOR =
1.87; 95% CI:1.68–2.08) and among female patients (aOR = 1.35; 95% CI: 1.27–1.42).
Patients with one or more mental health diagnoses had higher adjusted rates of hospital visits compared to those without any mental health diagnosis including addiction (aRR = 1.74,
95% CI: 1.58–1.91) and anxiety (aRR = 1.28, 95% CI: 1.23–1.32). 14.5% of patients with a
psychiatric diagnosis were referred to the hospital for specialized psychiatric services, and
38% of patients referred were eventually seen in consultation. The median wait time from
the date of referral to the date of consultation was 133 days.

Hospital. The data may contain patient-related information, and therefore due to privacy concerns, has not been made publicly available. However, the data will be made available upon request, and can be obtained by contacting the Office of Research and. Innovation: info.research.innovation@nygh.on.ca.

**Funding:** Support for this project was provided by a Health Databank Collaborative Grant from North York General Hospital, which provided in kind funding for biostatistician support to work on the study. Dr. Greiver is supported through the Gordon F. Cheesbrough Research Chair in Family and Community Medicine from North York General Hospital. The funder had no role in study design, data collection and analysis, or preparation of this manuscript.

**Competing interests:** The authors have declared that no competing interests exist.

## Conclusions

In this community, individuals with mental health diagnoses accessed primary and hospital-based health care at greater rates than those without mental health diagnoses. Wait times for specialized psychiatric care were long and most patients who were referred did not have a consultation. Information about services for patients with mental health conditions can be used to plan and monitor more integrated care across sectors, and ultimately improve outcomes.

## Introduction

One in five Canadians accesses services or supports for mental health concerns or addiction [1]. However, only a fraction of those suffering from mental health conditions or addiction seek care [2]. The Canadian Mental Health Association estimates that 49% of those with anxiety or depression have never sought help for their mental health issues [3]. Reasons for this include fear of judgment or stigma, not knowing who or where to ask for help, features of the mental illness itself (i.e. too depressed to leave one's house), and a lack of a perceived need for help [4]. Family physicians are often the first point of contact for patients with mental health concerns [5], and subsequently refer patients to mental health services if their needs cannot be managed in the primary care setting. However, in Canada, access to specialized mental health care services are limited [6], and there can be long wait lists for access to psychiatric care [7–9].

In 2006, the Canadian Psychiatric Association (CPA) published recommendations for wait time benchmarks for Canadians with serious psychiatric illness [10]. For example, the CPA suggests a patient with major depression of a non-urgent nature be seen by a psychiatrist within four weeks of referral. Existing data demonstrate that wait time averages for psychiatric services generally fall outside of the recommended timeframe [8, 9, 11]. The longer patients with more complex mental health needs (i.e. psychosis) wait for referral to mental health services, the greater the impact on certain health outcomes [12]. Those who wait longer for initial referral have a greater number and duration of inpatient admissions and increased impairment scores on validated questionnaires [12]. While there are known geographic differences in accessibility to mental health services, it is important to understand local patterns to guide system design and resource allocation [13]. Healthcare utilization is determined by multiple factors related to patient and health system characteristics. One common explanatory framework for this is the Andersen's behavioural model of health service use [14, 15]. This describes 'predisposing' factors such as demographics and health beliefs, 'enabling' factors such as financing and organization, and 'need' factors related to symptoms and functional impairment from disease.

Ontario, Canada has recently initiated a process of health system transformation through Ontario Health Teams (OHTs), an innovative model of health care delivery which puts the patient at the centre of an integrated system grounded in local partnerships and a commitment to local communities [16]. OHTs will have responsibility for the administration and delivery of health services to a defined population within a geographic region; services include those provided through hospital, primary care and home care settings. Improving mental health and addiction services is one of the key features of the OHT concept and is one of the three Year One priorities for the North York Toronto Health Partners (NYTHP) OHT, which is located in the northeast part of Toronto, Canada [17]. Locally and regionally relevant data are essential to plan and evaluate health system reform and implementation. This study will provide timely

information on regional access to mental health services from which the NYTHP OHT can build on.

Our study aims to identify patterns of health service use among people with mental health conditions and/or addictions in the North York community. While the results of this study will support care improvement plans for people with mental illness in the community being studied, similar processes could be replicated for integrated care in other OHTs across the province.

The objectives of this study were to: (i) determine patient characteristics associated with any diagnosis of a mental health condition, (ii) determine patient factors, including mental health conditions or addictions associated with increased health system utilization in our community; and (iii) describe wait times for referrals to specialized hospital-based mental health services among patients diagnosed with mental health conditions.

## Methods

This was a retrospective cohort study using linked primary care and hospital electronic data [18]. The Strengthening the Reporting of Observational Studies in Epidemiology (STROBE) guidelines were used to report the study [19].

### Data sources

This study utilizes data from the Health Databank Collaborative (HDC), a database containing linked health data derived from primary care and hospital clinical databases in North York [20]. The primary care data are collected from the Electronic Medical Records of patients registered to consenting physicians at the North York Family Health Team (NYFHT) [21], a primary care organization that provides community-based care to its patients. The NYFHT currently includes 91 family physicians and 40 allied health professionals. The HDC also includes data on hospital services for the same patients at North York General Hospital (NYGH), a community academic hospital. The HDC has previously been used to study health conditions across both hospital and primary care settings [18, 22], and offers an opportunity to support data-driven decision-making in mental health. The study timeframe was selected to include the maximum available follow up available in the HDC; the database includes data from January 1, 2012 –March 31, 2019.

### Settings and participants

Data were extracted and linked as of March 31st 2019. An initial cohort was generated by including patients aged 18 or older as of March 31st 2019. This was used to establish a second cohort by including patients with one or more mental health diagnoses identified in their record between January 1st 2012 and Mar 31st 2019. Patients were added to the second cohort as of the earliest available encounter record after January 1st 2012 indicating a mental health diagnosis in primary care or hospital data. Patients were censored on the administrative cut-off of study follow-up (i.e March 31, 2019), or on the last day of the calendar year if they were recorded as deceased.

Common mental health diagnoses were identified for inclusion, including: depression, anxiety, psychosis (including schizophrenia), addiction, and eating disorders. This approach builds on a previously developed approach for searching both primary care and hospital data [18]. For hospital records, the Canadian Institute for Health Information (CIHI)'s National Ambulatory Care Reporting System (NACRS; for emergency department data) [23] and Discharge Abstract Database (DAD; for inpatient admissions) [24] were used. The following ICD-10 codes were included: for psychosis F20, F25, F29; for bipolar disorder F30, F31; for

depression F32, F33, F34, F38, F39; for anxiety disorders (including generalized anxiety disorder, phobic disorders, obsessive compulsive disorder, agoraphobia, social phobia, panic disorder, post-traumatic stress disorder) F40-43; for addiction F10-F19; and for eating disorders F50. For primary care, patients with a documented diagnosis of bipolar disorder (ICD9 296 or billing code Q020), psychotic disorder (ICD9 codes 295, 298 or billing code Q021), depression (ICD9 code 309, 311), anxiety (ICD9 code 300), addiction (ICD9 code 303 or 304), eating disorder (ICD9 code 307) or free text indicating the diagnosis were included. The codes or free text must be present in the Health Conditions area of the summary health profile or in billing data. The search criteria are provided in S1 Appendix.

### Variables

A hospital visit was defined as an encounter related to an Emergency Department (ED) visit, inpatient medical or surgical hospital admission, or inpatient psychiatric admission. An outpatient mental health visit (i.e. a scheduled, non-urgent appointment with psychiatry) was not defined as a hospital visit. Primary care visits were defined using the appropriate list of billing codes [25].

The patient's age (as of March 31$^{st}$ 2019) and sex were obtained from the HDC. Socioeconomic quintiles were derived using geographical information based on Statistics Canada's Postal Code Conversion File [26]. Rural neighbourhoods were identified using the rural postal codes with "zero" in the second position of the postal codes [26–28]. Deceased status was derived using demographic data in the HDC. Comorbidities were identified using primary care data with previously validated definitions for the following chronic conditions: diabetes, hypertension, osteoarthritis, COPD, dementia, epilepsy and Parkinson's Disease [29].

### Statistical methods

A negative-binomial regression model was used to accommodate for overdispersion of main outcomes: (a) hospital visits; (b) primary care visits; (c) either hospital or primary care visits. The rate ratios were used to compare the hospital and/or primary care visits (a-c) with respect to the following patient characteristics: age, sex, income quintile, rurality, depression, anxiety, psychosis, addiction, and eating disorders and deceased status.

Secondary analyses were peformed for the presence of any mental health diagnosis defined as the onset of at least one of the following conditions during the study period: depression, anxiety, psychosis, addiction, or eating disorders. A logistic regression model was used to estimate the odds ratios with respect to patient characteristics associated with at least one diagnosis of mental health condition in either primary care or hospital.

Wait time estimates among patients with mental health conditions who were referred to outpatient psychiatry from primary care or hospital settings were identified. Wait times were defined as the difference between the earliest referral date and the visit date for the outpatient mental health visit at the hospital within the study follow-up. A look-back window of twelve months with respect to earliest referral date was implemented to account for possibility in which a hospital outpatient mental health visit preceded the referral date.

Analyses were performed using SAS v9.4 M6. The North York General Hospital Research Ethics Board approved the study.

## Results

A flowchart for the cohort generation is shown in Fig 1. There were 28,482 adult patients in the HDC (as of March 31, 2019) and 8,559 patients (30.1% of adults) had at least one mental health condition in their record during the study period.

**Fig 1. Primary cohort generation (step 1–2, general adult population) and secondary cohort generation (step 3, patients with a mental health condition) in the Health Databank Collaborative.**

Table 1 shows overall healthcare utilization in primary care and utilization of mental health services amongst all adult patients in the hospital. Using the cohort of 28,482 adult patients, there were 24,764 patients (86.9%) with one or more primary care visits; 19,403 patients (68.1%) with an ED visit; 8,967 patients (31.5%) with an inpatient medical or surgical admission; 121 patients (0.5%) with a psychiatric hospital admission; 1560 patients (5.5%) with one or more outpatient mental health visit(s) during the study follow-up period from January 1st 2012 to March 31st 2019.

Fig 2 shows adjusted rate ratios (aRR) of healthcare utilization in primary care and hospital. The adjusted rate of hospital visits was significantly higher among patients with some mental health conditions than among those without the condition. These conditions included addiction (aRR = 1.74, 95% CI: 1.58–1.91), anxiety (aRR = 1.28, 95% CI: 1.23–1.32), depression (aRR = 1.06, 95% CI: 1.01–1.10), eating disorder (aRR = 1.08, 95% CI: 1.01–1.15). The adjusted rate of primary care visits was higher among those with anxiety (aRR = 1.65, 95% CI: 1.60–1.70), depression (aRR = 1.17, 95% CI: 1.13–1.20), eating disorder (aRR = 1.48, 95% CI: 1.39–1.55), and psychosis (aRR = 1.18, 95% CI: 1.01–1.38).

Fig 3 shows adjusted odds ratios (aOR) for the characteristics of patients diagnosed with any mental health condition. Younger patients had greater aOR of a diagnosis of any mental health condition (18–30 years vs. 81+years aOR = 1.87; 95% CI:1.68–2.08) as did female patients (aOR = 1.35; 95% CI: 1.27–1.42). There was no association between a diagnosis of a mental health condition and neighbourhood income quintile.

1,239 of 8,559 patients with a mental health condition (14.5% of the total cohort) were referred to outpatient Psychiatry at NYGH. 466 (37.6% of those referred) had an outpatient

**Table 1. Health care utilization among patients in the Health Databank Collaborative from January 1st 2012 to March 31st 2019.**

| Patient characteristics | At least one Primary care visit | | At least one Emergency Department visit | | At least one Inpatient visit | | At least one Psychiatric admission | | At least one Outpatient mental health visit | | Total |
|---|---|---|---|---|---|---|---|---|---|---|---|
| | N | Row percent (%) | N | Row percent (%) | N | Row percent (%) | N | Row percent (%) | N | Row percent (%) | N |
| **Age group** | 2555 | 88.3% | 2437 | 84.2% | 442 | 15.3% | 16 | 0.6% | 350 | 12.1% | 2894 |
| **18–30 years** | | | | | | | | | | | |
| **31–40 years** | 3557 | 87.0% | 2632 | 64.4% | 1895 | 46.4% | 11 | 0.3% | 227 | 5.6% | 4088 |
| **41–50 years** | 3577 | 85.3% | 2709 | 64.6% | 1149 | 27.4% | 10 | 0.2% | 246 | 5.9% | 4194 |
| **51–60 years** | 4015 | 90.3% | 2665 | 60.0% | 594 | 13.4% | 15 | 0.3% | 246 | 5.5% | 4445 |
| **61–70 years** | 3837 | 91.1% | 2450 | 58.2% | 848 | 20.1% | 12 | 0.3% | 230 | 5.5% | 4212 |
| **71–80 years** | 3368 | 89.8% | 2373 | 63.3% | 1161 | 31.0% | 19 | 0.5% | 123 | 3.3% | 3751 |
| **81+ years** | 3855 | 78.7% | 4137 | 84.5% | 2878 | 58.8% | 38 | 0.8% | 138 | 2.8% | 4898 |
| **Sex** | 16563 | 86.8% | 12753 | 66.8% | 6703 | 35.1% | 80 | 0.4% | 1073 | 5.6% | 19080 |
| **F** | | | | | | | | | | | |
| **M** | 8201 | 87.2% | 6650 | 70.7% | 2264 | 24.1% | 41 | 0.4% | 487 | 5.2% | 9402 |
| **Income Quintiles** | 906 | 43.6% | 1658 | 79.7% | 1200 | 57.7% | 23 | 1.1% | 135 | 6.5% | 2079 |
| **1 (= lowest)** | 4159 | 88.6% | 3489 | 74.4% | 1677 | 35.7% | 21 | 0.4% | 307 | 6.5% | 4692 |
| **2** | 4485 | 90.3% | 3563 | 71.7% | 1636 | 32.9% | 28 | 0.6% | 294 | 5.9% | 4968 |
| **3** | 4162 | 90.4% | 3149 | 68.4% | 1442 | 31.3% | 17 | 0.4% | 253 | 5.5% | 4602 |
| **4** | 3625 | 90.7% | 2387 | 59.7% | 1127 | 28.2% | 6 | 0.2% | 162 | 4.1% | 3996 |
| **5 (= highest)** | 7427 | 91.2% | 5157 | 63.3% | 1885 | 23.1% | 26 | 0.3% | 409 | 5.0% | 8145 |
| **Region** | 275 | 85.9% | 118 | 36.9% | 80 | 25.0% | * | * | 11 | 3.4% | 320 |
| **Rural** | | | | | | | | | | | |
| **Urban** | 24489 | 87.0% | 19285 | 68.5% | 8887 | 31.6% | 119 | 0.4% | 1549 | 5.5% | 28162 |
| **Total** | 24764 | 86.9% | 19403 | 68.1% | 8967 | 31.5% | 121 | 0.4% | 1560 | 5.5% | 28482 |

*suppressed due to cell count <5.

Psychiatry visit at NYGH during the study period. The remaining 773 patients (62%) who were referred did not have an outpatient Psychiatry visit at NYGH by the end of the study follow-up (i.e. March 31, 2019); these patients were administratively censored at the end of the study follow-up (i.e. March 31, 2019). The median wait times from the referral date to outpatient Psychiatry appointment date was 133 days (IQR: 48 days– 317 days).

## Discussion

In this community-based sample of patients in Toronto, Canada, those with a mental health diagnosis were more likely to be young and female, and had more healthcare visits to their family physician or their hospital than those without a mental health diagnosis. Addictions were associated with higher rates of hospital visits, while a diagnosis of anxiety was associated with more primary care visits. Almost two thirds of those referred to the hospital's outpatient Psychiatry department were not seen, and the median wait time for those seen was over four months.

In contrast to other studies [30], patients living in neighbourhoods at lower income quintiles were not more likely to have a mental health diagnosis. The reasons for this finding are not obvious; it is possible that this finding may not accurately represent diagnosis but rather service use, and although lower socioeconomic status (SES) is consistently associated with higher rates of mental illness, it is not necessarily associated with greater mental health care service utilization when adjusting for other patient characteristics. It is well known that persons living with poverty encounter many barriers to obtaining health care [31]. It is possible

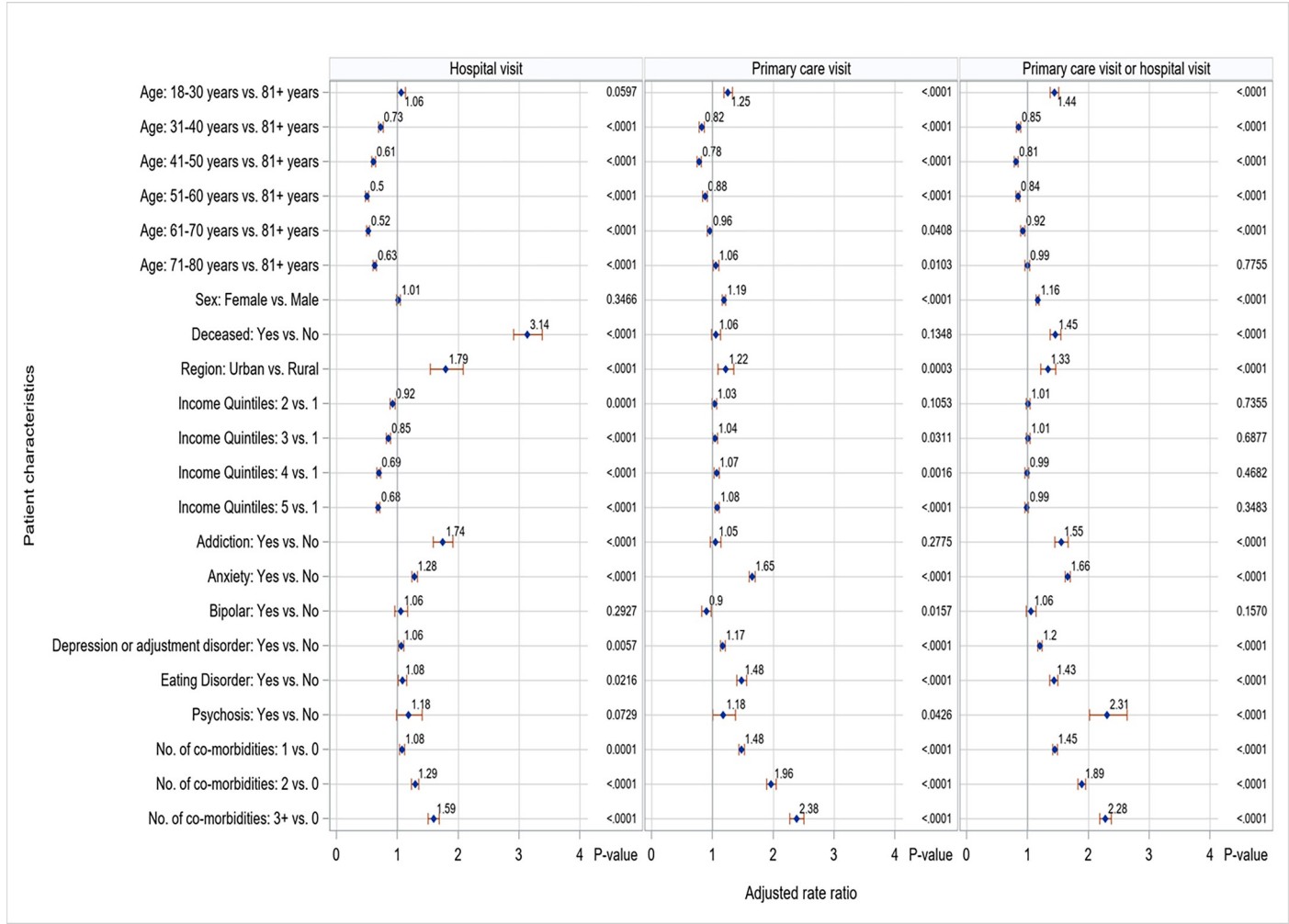

**Fig 2. Adjusted rate ratios for health care utilization in hospital and primary care.**

that some individuals who are of a lower income quintile and are facing mental health challenges may find it difficult to enroll with a primary care physician and are therefore not sampled in our study. North York is made up of higher socioeconomic households compared to many other neighbourhoods in Toronto [32]. While the group in our study with a mental health diagnosis may not be representative of Canada as a whole, it may be representative of other Canadian settings with similar culturally-diverse populations, or even other countries with similar ethnocultural makeups.

Most patients (83.5%) with mental health diagnoses were not referred for outpatient psychiatric assessment during the study period. It is likely that most patients with mental health diagnoses were managed exclusively in the primary care setting, and therefore referral was not deemed necessary. Referral rates seen in this study are in line with other estimates of referral rates for mental health services (11–36%) [33]. Median wait time for referral to outpatient psychiatry was 133 days. These findings are in keeping with studies by Jaakkimainen et al. (2014) and Liddy et al. (2020) who reported that Psychiatry had one of the longest wait times for specialist referral, with median wait times of 73 and 88 days, and 75th centile waits of 231.5 days and 233 days, respectively.

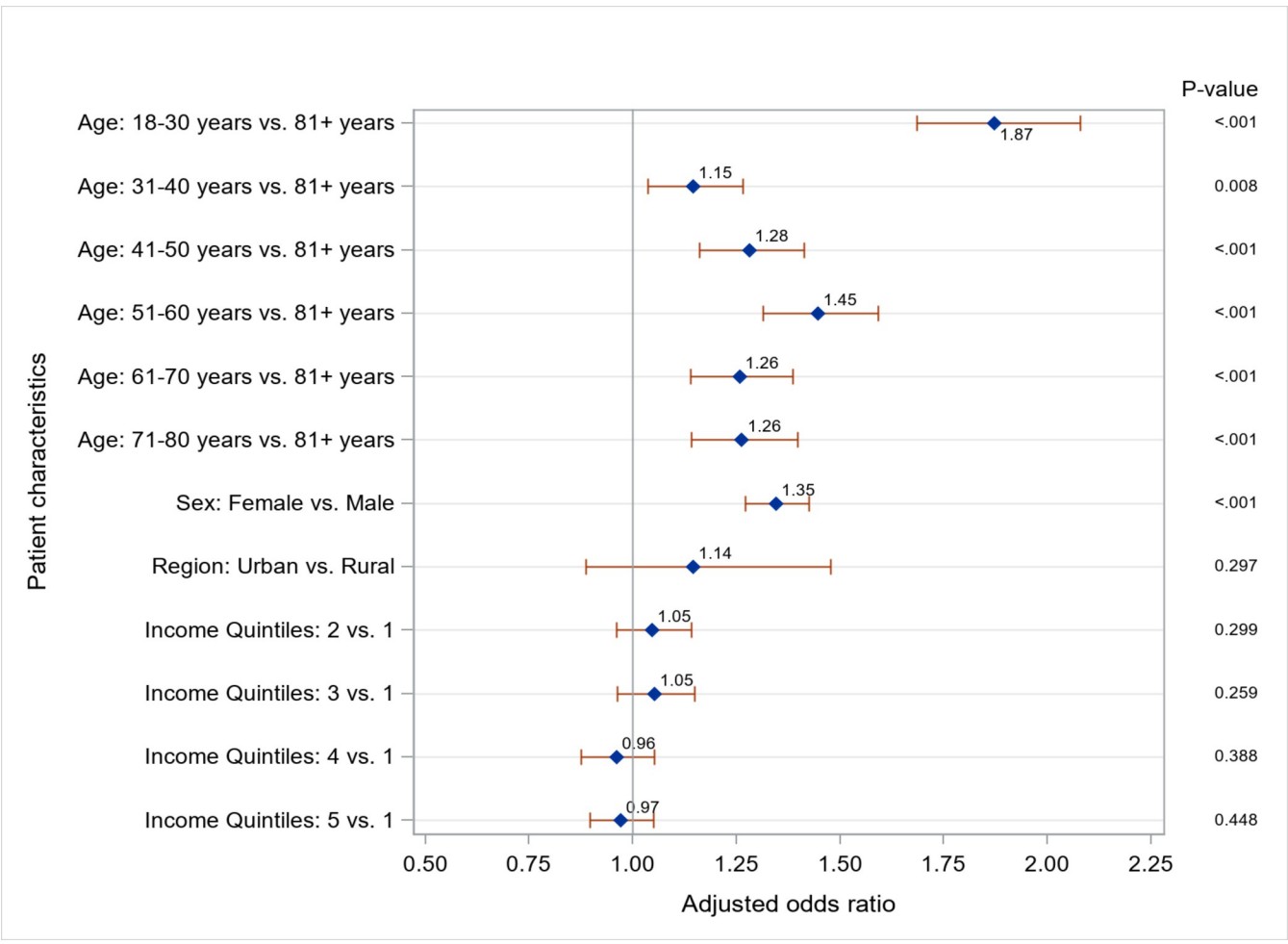

**Fig 3. Adjusted odds ratios for a diagnosis of mental health conditions.**

For the majority of patients who were not seen despite being referred during the study period, there are multiple possible explanations: they may have not attended the appointment; symptoms may have resolved by the time of the appointment and therefore no longer needed to be seen; they may have been referred during the study period but their appointment was booked after the study period ended; they may have been referred to another service concurrently and received assessment there. There are other mental health assessment services available in the Greater Toronto Area where patients may have been seen and those data are not included in the current study. Furthermore, when referrals are triaged by hospital staff, patients are frequently provided appropriate resources. Data about whether patients accessed those (such as counselling services or information about how to access private psychotherapy), were not available for this study. These findings about the rates of use of specialty mental health care are similar to studies in other locations. For example, a study about health services use among people with mental health diagnoses in several large cities in the United States found about 20% of people accessed mental health specialty care over a several year period [34]. A similar proportion of use of specialized mental health services was identified among people with depression in Finland [35]. There are fewer descriptions of wait times for similar

services in the existing literature: for example, a study in a US community health centre (primary care) setting described a wait time of 13 days for a psychiatric appointment [36].

There are a number of study limitations. Firstly, the list of mental health diagnoses included in this study are not exhaustive in the inclusion of all mental health diagnoses (such as, for example, personality disorders). Additionally, because the database is de-identified, we were unable to perform chart audits to confirm accuracy of diagnoses. Inclusion and labeling is dependent on accurate diagnosis and chart labeling by the family physician or hospital physician, although we note that the hospital data are assessed by trained abstractors [23, 24]. We did not report some specific diagnoses (i.e. obsessive-compulsive disorder) but rather grouped all disorders together under one heading (i.e. anxiety disorders). This limits our ability to make conclusions about specific diagnoses, and there may be a large difference in severity of illness, referral patterns, etc. for the individual diagnoses under the broader heading which are not being captured by this study. From a health services research perspective, we were somewhat limited in addressing all components of Andersen's model by the data to which we had access: the underlying database did not include some data elements that would be necessary to characterize healthcare utilization according to all of Andersen's concepts, such as race and ethnicity and symptom severity. However, we were able to assess aspects of all of those key factors. We chose to combine the diagnoses of 'depression' with 'adjustment disorder', as many primary care providers might use the billing code for 'adjustment disorder' synonymously with the billing code for 'depression', even though they are two separate billing codes. This may have overestimated the true 'depressed' population. There are also concerns about the validity of a diagnosis of adjustment disorder which may overestimate the number of patients in this group. Some patients may also choose not to disclose their mental health concerns to their physician leading to incomplete or inaccurate mental health diagnoses, and therefore would not be identified in available data as receiving care from either their primary care provider or hospital.

13.1% of patients in HDC did not have a primary care visit during the study period. The reason for this could be that the primary care visit falls outside the study period or that there is no billing data for the primary care visit, as primary care visit is defined using billing records in this study. However, we think this is relatively unlikely since everyone included in the study had publicly-funded provincial health care coverage and therefore all visits would be eligible for billing to the Ontario Ministry of Health. Since the study population in this article was limited to localized primary care offices and a single community hospital, it is necessary to acknowledge the limitations in generalizability to the Canadian population, and elsewhere. This was a retrospective study with a non-random (i.e., convenience) sampling frame of patients and providers, and therefore, the findings may not be generalizable externally [37].

We did not have access to the specific details of inpatient mental health admissions which is captured in the Ontario Mental Health Reporting System (OMHRS) [38] which would have added additional information about severity of illness. Furthermore, data from private mental health services (such as psychologists and psychotherapists) paid for by patients out of pocket or through private insurance were unavailable.

Future research exploring the referral patterns from primary to secondary mental health services could look at the reason for referral i.e. diagnostic clarification vs. medication management vs. provision of therapy, to allow for better resource planning.

There are many potential health system implications from these findings to better service patients with mental health diagnoses, including initiatives that are already underway locally, provincially and nationally. During the COVID-19 pandemic, the Canadian government has provided additional funding for digital mental health services; the continuation of these services may be one promising approach for increasing access to outpatient services. There is

already a larger system movement towards a Stepped Care Approach [39], a system of delivering and monitoring mental health treatment so that the most effective and evidence-based, yet least resource intensive treatment, is delivered first, which is consistent with the NYTHP OHT's Year One priority to link patients to evidence-based treatments (i.e. Cognitive Behavioral Therapy) through an OHT centralized intake [17].

Although there are some limitations to generalizability, there are broader lessons to be learned from this study that can be passed on to non-Canadian readers. This study highlights high need but poor uptake of specialized mental health services. In order to deliver care for the right individuals in the right place at the right time, the siloed nature of mental health care needs to be broken, and more linear pathways to access care in a step-wise approach should be undertaken. Improved data sharing between primary and secondary centres is a necessary first step.

A greater understanding of which clinical or demographic characteristics in a jurisdiction are most predictive of high service use will support the development of improved triage systems for specialized mental health care service planning and delivery.

This study identified higher primary and secondary health care utilization among those with a mental health diagnosis in a large cohort of adult patients from Toronto, Canada, as well as long wait times and high rates of loss of follow-up among people referred for hospital-based outpatient mental health services. By understanding which patient characteristics predict symptom severity and health service utilization, we will hopefully be better able to identify system gaps and prioritize early referral and added resource allocation for more severe cases, ultimately improving treatment outcomes and reducing wait times and cost to the health care system.

## Supporting information

**S1 Appendix. Search criteria for mental health conditions in health databank collaborative.**
(DOCX)

## Acknowledgments

The authors are grateful to the physicians and patients who allow data sharing to make the HDC possible.

## Author Contributions

**Conceptualization:** Kimberly Lazare, David Koczerginski, Maria Muraca, Wai Lun Alan Fung, Braden O'Neill.

**Data curation:** Babak Aliarzadeh, Steven Bernard, Rahim Moineddin.

**Formal analysis:** Sumeet Kalia.

**Methodology:** Kimberly Lazare, Sumeet Kalia, Babak Aliarzadeh.

**Project administration:** Kimberly Lazare.

**Writing – original draft:** Kimberly Lazare, Sumeet Kalia, Braden O'Neill.

**Writing – review & editing:** Kimberly Lazare, Sumeet Kalia, Babak Aliarzadeh, Steven Bernard, David Eisen, Michelle Greiver, David Kaplan, David Koczerginski, Maria Muraca, Wai Lun Alan Fung, Braden O'Neill.

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
