## [Decision Letter · Decision Letter 0]

11 Nov 2021

PONE-D-21-14663Health system use among patients with mental health conditions in a community based sample in Toronto, Canada: A retrospective cohort studyPLOS ONE

Dear Dr. Lazare,

Thank you for submitting your manuscript to PLOS ONE. After careful consideration, we feel that it has merit but does not fully meet PLOS ONE’s publication criteria as it currently stands. Therefore, we invite you to submit a revised version of the manuscript that addresses the points raised during the review process.

We look forward to receiving your revised manuscript.

Kind regards,

Enkeleint A. Mechili

Academic Editor

PLOS ONE

Journal Requirements:

https://journals.plos.org/plosone/s/file?id=wjVg/PLOSOne_formatting_sample_main_body.pdf and https://journals.plos.org/plosone/s/file?id=ba62/PLOSOne_formatting_sample_title_authors_affiliations.pdf,

Reviewers' comments:

Reviewer's Responses to Questions

**Comments to the Author**

1. Is the manuscript technically sound, and do the data support the conclusions?

Reviewer #1: Yes

Reviewer #2: Yes

2. Has the statistical analysis been performed appropriately and rigorously? 

Reviewer #1: Yes

Reviewer #2: Yes

3. Have the authors made all data underlying the findings in their manuscript fully available?

Reviewer #1: Yes

Reviewer #2: Yes

4. Is the manuscript presented in an intelligible fashion and written in standard English?

Reviewer #1: Yes

Reviewer #2: Yes

5. Review Comments to the Author

Reviewer #1: This Canadian paper is well written and the authors clearly mention the limitations of this cross-sectional study.

I have two remarks; the first one is about the lack of a theoretical framework that has guided the study. In the literature, there are several theoretical models that address utilization care services. One of them is the Andersen health behavior model that has been widely recognized as a reliable tool for the study of health services utilization. According to this model, health service utilization could be considered as function of three sets of factors: predisposing (demographic and social) factors, enabling (economic) factors, and need (health outcomes) factors. The authors could address adequately the theoretical reasoning of their study and explain to what extent it guided the study design.

The second one is relevant with the external validity of the study. It is obvious that the study findings cannot be generalized in setting outside the Canadian context. However, there are issues that can be learned and transmitted to the non-Canadian readers and the authors could discuss further this subject in a revised version.

Reviewer #2: General comment: The manuscript reported health service use among people identified as having mental health conditions, which is very important for psychiatric patients as access to health care facilities is so crucial issue.

Keywords: Much better to have keywords as Mesh term to make your research more popular. Please review and amend accordingly

Abstract: Try to avoid first person pronouns when writing academic papers. I would suggest reviewing the method section in the abstract.

Introduction: is well-written and clear to me.

Methodology: The Methods section provided in depth details. However, authors did not explain why they have selected such duration (2012-2019), was that based on a specific factor? Also, regarding common mental health diagnosis: why dual diagnosis was not part in this?

Results: No comments, the results section is thorough and supported by the analysed data.

Discussion: The Discussion section is comprehensive. However, I was hoping to see more of other comparative trials from other places in the world when comparing the results to make sense of the reproducibility of the research.

Conclusion: well-written

6. PLOS authors have the option to publish the peer review history of their article (what does this mean?). If published, this will include your full peer review and any attached files.

Reviewer #1: No

Reviewer #2: No

---

## [Author Response · Author response to Decision Letter 0]

12 Feb 2022

November 18, 2021

Rebuttal Letter: PONE-D-21-14663

Dear Editors,

We thank the reviewers for their thoughtful comments regarding our manuscript entitled Health system use among patients with mental health conditions in a community-based sample in Toronto, Canada: A retrospective cohort study. We have edited the manuscript to address their concerns. We believe that the manuscript is now suitable for publication in PLOS One.

Reviewer #1: This Canadian paper is well written and the authors clearly mention the limitations of this cross-sectional study. I have two remarks; the first one is about the lack of a theoretical framework that has guided the study. In the literature, there are several theoretical models that address utilization care services. One of them is the Andersen health behavior model that has been widely recognized as a reliable tool for the study of health services utilization. According to this model, health service utilization could be considered as function of three sets of factors: predisposing (demographic and social) factors, enabling (economic) factors, and need (health outcomes) factors. The authors could address adequately the theoretical reasoning of their study and explain to what extent it guided the study design. 

Thank you for the suggestion to enhance our description of the theoretical reasoning of our study. We have included details about the ‘Andersen’ model and how it relates to what data we had access to as follows: 

In ‘introduction’ (p.2): “Healthcare utilization is determined by multiple factors related to patient and health system characteristics. One common explanatory framework for this is the Andersen’s behavioural model of health service use [14]. This describes ‘predisposing’ factors such as demographics and health beliefs, ‘enabling’ factors such as financing and organization, and ‘need’ factors related to symptoms and functional impairment from disease.”

In ‘interpretation’ (p.13): “In this study we were limited in addressing all components of Andersen’s model by the data to which we had access: the underlying database did not include some data elements that would be necessary to characterize healthcare utilization according to all of Andersen’s concepts, such as race and ethnicity and symptom severity. However, we were able to assess aspects of all of those key factors.”

The second one is relevant with the external validity of the study. It is obvious that the study findings cannot be generalized in setting outside the Canadian context. However, there are issues that can be learned and transmitted to the non-Canadian readers and the authors could discuss further this subject in a revised version.

Thank you for this important point, which we have further addressed in the manuscript as follows (interpretation, p.14): 

“Since the study population in this article was limited to localized primary care offices and a single community hospital, it is necessary to acknowledge the limitations in generalizability to Canadian population, and elsewhere. This was a retrospective study with non-random (i.e., convenience) sampling frame of patients and providers, and thereby the findings may not be generalizable externally (Jager et al, 2017).” 

That being said, to your point we have gone ahead and included in our Discussion section what can be learned from this study and transmitted to non-Canadian readers.

Reviewer #2: 

Keywords: Much better to have keywords as Mesh term to make your research more popular. Please review and amend accordingly.

Thank you. We have adjusted the keywords to be Mesh terms as per your suggestion. The new keywords are: Health Services, Primary Health Care, Referral and Consultation, Mental Health Services, Community Mental Health Services, and Cohort Studies.

Abstract: Try to avoid first person pronouns when writing academic papers. I would suggest reviewing the method section in the abstract.

Thank you. We have removed first person pronouns throughout the manuscript.

Methodology: The Methods section provided in depth details. However, authors did not explain why they have selected such duration (2012-2019), was that based on a specific factor? 

We have been more specific about this in the manuscript as follows (methods, p.5): “The study timeframe was selected to include the maximum available follow up available in the HDC; the database includes data from January 1, 2012 – March 31, 2019.”

Also, regarding common mental health diagnosis: why dual diagnosis was not part in this?

Two different meanings of the term ‘dual diagnosis’ exist. In Canada, dual diagnosis often refers to the coexistence of one or more mental health problem with a developmental disability. In other jurisdictions such as the United States of America, dual diagnosis might refer to a psychiatric disorder along with, for example, a substance use issue. We interpreted your question about dual diagnosis to refer to the former which is applicable to the Canadian context.

The reason for not including dual diagnosis in our list of common mental health diagnoses was because NYGH does not have a dual diagnosis clinic, and many other tertiary centres in the Greater Toronto Area (GTA) do, such as the Centre for Addiction and Mental Health (CAMH), Ontario Shores Centre for Mental Health Sciences and Surrey Place. Based on the experience of clinicians on our team we assessed that patients with dual diagnosis may already have a connection to one of these other hospitals, and would instead access acute medical or psychiatric care at those other hospitals.

Discussion: The Discussion section is comprehensive. However, I was hoping to see more of other comparative trials from other places in the world when comparing the results to make sense of the reproducibility of the research.

Thank you, we have provided additional details about related studies on this issue as follows (interpretation, p. 12): These findings about the use of specialty mental health care are similar to studies in other locations. For example, a study about health services use among people with mental health diagnoses in several large cities in the United States found about 20% of people accessed mental health specialty care over a several year period. not accessing specialty mental health care [33]. A similar proportion of use of specialized mental health services was identified among people with depression in Finland [34]. There are fewer descriptions of wait times for similar services in the existing literature, but available descriptions vary widely: a study in a US community health centre (primary care) setting described a wait time of 13 days for a psychiatric appointment [35].

Sincerely,

Kimberly Lazare, MD, CCFP, MScCH

Corresponding Author

---

## [Editor Report · Decision Letter 1]

21 Mar 2022

Health system use among patients with mental health conditions in a community based sample in Toronto, Canada: A retrospective cohort study

PONE-D-21-14663R1

Dear Dr. Lazare,

We’re pleased to inform you that your manuscript has been judged scientifically suitable for publication and will be formally accepted for publication once it meets all outstanding technical requirements.

Kind regards,

Enkeleint A. Mechili

Academic Editor

PLOS ONE
---

## [Editor Report · Acceptance letter]

28 Apr 2022

PONE-D-21-14663R1 

Health system use among patients with mental health conditions in a community based sample in Toronto, Canada: A retrospective cohort study 

Dear Dr. Lazare:

I'm pleased to inform you that your manuscript has been deemed suitable for publication in PLOS ONE. Congratulations! Your manuscript is now with our production department. 

Kind regards, 

on behalf of

Dr. Enkeleint A. Mechili 

Academic Editor

PLOS ONE